# Preventable Disease, the Case of Colorado: School District Demographics and Childhood Immunizations

**DOI:** 10.3390/vaccines10101579

**Published:** 2022-09-21

**Authors:** Katherine Nicolich, Jacob Gerken, Blaire Mallahan, David W. Ross, Isain Zapata

**Affiliations:** 1Division of Clinical Medicine and Surgery, Rocky Vista University, Parker, CO 80112, USA; 2Department of Biomedical Sciences, Rocky Vista University, Parker, CO 80112, USA; 3Department of Pediatrics, Oregon Health & Science University, Portland, OR 97239, USA

**Keywords:** vaccine hesitancy, vaccine exemptions, poverty, racial and cultural

## Abstract

The objective of this study was to evaluate the impact of selected sociodemographic factors on childhood vaccination hesitancy and to define their role according to specific exemptions. This population-based cohort study utilized vaccination rate and sociodemographic data from 1st to 12th grade from 2017 to 2021 for all school districts in Colorado. Data included immunization status and exemptions for each vaccine, race, ethnicity, and free and reduced meal status. Data were evaluated through dimensional analysis and Generalized Linear Mixed Models. School districts with a higher representation of White students had lower immunization rates and use more personal exemptions while school districts with larger Hispanic populations and higher proportions of free and reduced lunches had higher vaccination rates and used more religious exemptions. Black and Pacific Islander populations had higher rates of incomplete vaccination records while Asian American population displayed increased vaccination compliance. Colorado is a robust example of how socioeconomic and cultural differences are important factors with a direct influence on vaccination rates. Future childhood vaccination campaigns and legislation should consider complex socioeconomic and cultural factors.

## 1. Introduction

Vaccination against communicable diseases is essential to preventing mortality and morbidity in children [1]. Due to various socioeconomic and political factors, as well as the COVID-19 pandemic, childhood vaccination rates across the United States decreased throughout 2020–2021 [2,3]. In 2017, Colorado ranked 43rd out of 50 states with only 64% of children immunized at 24 months of age with the 7-vaccine series recommended by the Centers for Disease Control (CDC). This rate improves at 36 months of age with 78% of children immunized [4]. The World Health Organization Strategic Advisory Group of Experts (WHO-SAGE) defines vaccine hesitancy as a behavior that leads a “delay in acceptance or refusal of vaccines despite the availability of vaccine services” [5]. Hesitancy toward routine childhood vaccinations has played a role in outbreaks of measles, mumps, and pertussis around the United States [6,7]. Parents may refuse some or all recommended vaccinations for their children for many reasons. Vaccine hesitancy can be related to three main factors: complacency, convenience, and confidence. Complacency describes parents who do not see a need for all or some vaccinations. Convenience refers to access to a vaccine, including geographic access to a provider. Confidence refers to parents’ trust in a vaccine and its components [8].

Causes or antecedents to vaccine hesitancy are multifactorial and hinge on parental attitudes and beliefs [9,10]; however, they lead to a simple behavior of not accepting vaccines. Educational attainment of the parent has shown to be a strong predictor of children’s vaccination status [11,12,13,14]. Even though Colorado is a top state in educational attainment with 40.9% of Colorado adults over 25 years old having a bachelor’s degree, it still lags in childhood vaccination rates [4]. Racial differences have also been associated with vaccine hesitancy [15,16,17]. These racial differences are intimately intertwined with personal beliefs and other socioeconomic factors such as poverty that may affect confidence in vaccinations [16,18].

In addition, the COVID-19 pandemic resulted in a significant decrease in childhood vaccination rates in Colorado. After social distancing guidelines were implemented in Colorado on 15 March 2020, the rate of childhood immunization administration dropped 78% and 82% for children aged 3 to 9 years old and 10 to 17 years old, respectively from January to March 2020 [19]. The already low vaccination rates, compounded with the recent pandemic, highlights the importance of understanding different factors affecting childhood vaccination. Many of these concerns are reflected in COVID-19 vaccination rates [20]. Because of the unique social and demographic factors found in Colorado, the objective of this study was to evaluate the effect of some sociodemographic factors (race, ethnicity, and poverty) to childhood vaccination hesitancy behaviors and to evaluate the type of exemptions used at school to justify their decisions. This study provides important insight that can be useful in the development of more effective initiatives to mitigate vaccination hesitancy.

## 2. Materials and Methods

### 2.1. Vaccination Status and Demographic Factor Data

This cohort study was designed to evaluate and identify trends in vaccination rates and associations with sociodemographic factors from public data in Colorado. Vaccination rates for the Diphtheria, Tetanus and Pertussis (DTaP), Hepatitis B (HepB), Measles, Mumps and Rubella (MMR), Polio, Tetanus, Diphtheria and Pertussis (Tdap), and Varicella vaccines for each school district in Colorado was acquired for the 2017–2018, 2018–2019, 2019–2020, and 2020–2021 academic years from the Colorado Department of Public Health and Environment [21]. The data provided by the Colorado Department of Public Health and Environment separated vaccination status into fully immunized, in process, incomplete records, no records, medical exemptions, personal exemptions, and religious exemptions. The percentage of students who receive Free and Reduced meals and the ethnic makeup of each school district was obtained from the Colorado Department of Education [22]. This eligibility is determined by guidelines set and adjusted every year that take into account the household size and income. In 2022, a married couple with two children will require reporting less than $36,075 or less than $51,338 to qualify for free or reduced meals, respectively [23]. The proportion of students receiving free meals, reduced meals, and the combination of free and reduced meals together were evaluated separately. Only data from first grade to twelfth grade were included in the analysis. Data from Pre-school and kindergarten level was excluded in the study because of inconsistent reporting and low sample size. Databases included the values for total enrollment and the number of facilities in the school district. This study was categorized as exempt from the Institutional Review Board as the data is publicly available and provided in a deidentified format by school district with no unique identifiers. All data was collected as aggregate values per vaccine type per school year for each school district in the state of Colorado and assembled in a single dataset.

### 2.2. Statistical Analysis

Pearson correlation and factor analysis were performed on the vaccination status compliance and sociodemographic factors (race and poverty) from the full dataset that included all school district and all years. Factor analysis was performed using the principal component method on the correlation matrix. An Oblimin rotation was performed on the factors retained on the MinEigen criterion (eigenvalues greater than 1). Correlation and factor analysis were performed using PROC CORR and PROC FACTOR in SAS/STAT v.9.4 (SAS Institute Inc., Cary, NC, USA). Associations between individual compliance parameters (different levels of vaccine adoption, dependent variable) and sociodemographic factors (race and poverty; independent variables) were evaluated using Generalized Linear Mixed Models (GLMMs). The total enrollment for the year and the number of facilities were incorporated as covariates to address the effect of school district size and infrastructure. The effect of the individual school district was defined as a random effect. Residuals from all models were assumed to be normally distributed. All modeling estimation was performed using PROC MIXED and all descriptive statistics were calculated using PROC MEANS in SAS/STAT v.9.4. Significance is declared and presented for this study in two ways: as a *p* ≤ 0.05 threshold and as Bonferroni adjusted threshold (the numbers of tests for this adjustment are specified for each set). Exact *p* values are provided for all effects tested that are presented.

## 3. Results

The demographic makeup of the 178 Colorado school districts used in the analysis is detailed in Table 1. On average, 46.3% of Colorado K-12 students receive either free or reduced lunch. The demographic makeup of school districts in Colorado is predominantly White (64.6%) and Hispanic (29.1%), with the remaining 6.3% consisting of other minorities. The average student population per school district was 4412 students consisting of 9.28 facilities or distinct schools. Compliance and exemption type rate by vaccine rates are presented in Appendix A.

The association of vaccination status and vaccination exceptions to poverty indicators of the percentage of students receiving free meals, reduced meals, and a combined free and reduced meal category were evaluated through dimensional analysis and GLMMs. Due to the type of data, our assessments are not indicative of causality.

Correlation analysis and factor analysis of vaccination hesitancy and poverty and race indicators in Colorado are presented in Figure 1. Correlation analysis (Figure 1A) shows strong associations between poverty and racial indicators. One of the strongest effects seen was that school districts with larger Hispanic populations were negatively correlated with personal exceptions; while school districts with larger White populations were positively correlated with personal exemptions. Other correlations are strong but can be difficult to interpret due to the pairwise nature of correlation tests. For this reason, factor analysis (Figure 1B) is more useful and displays a fuller picture. Factor analysis on the rotated pattern of the first two dimensions show that demographic indicators align on the horizontal axis while vaccination record completeness aligns on the vertical axis with their exceptions running closer to the center on both axes. Race indicators show districts with larger White or districts with larger Hispanic populations in the extremes of the horizontal axis with other ethnic groups in the middle. All 3 poverty indicators overlap directly on top of Hispanics suggesting that this group is the most economically vulnerable in the state of Colorado in comparison to other groups. Personal exemptions lean towards districts with larger White populations while religious exemptions lean more toward districts with larger Hispanic populations. On the vertical axis record compliant-incomplete runs on its axis. Some racial indicators lean towards these ends where Black, Pacific Islander populations lean towards having incomplete records while Asian American populations lean towards compliance. These findings suggest that demographic disparities and cultural differences are strongly reflective of vaccination hesitancy patterns.

The subjective evaluations done by correlation and factor analysis can be further evaluated through a modeling approach for a more objective interpretation. Effect estimates and *p*-values for the poverty indicators modeled are presented in Table 2. School districts with a higher percentage of students receiving free meals had increased rates of full immunization (*p* = 0.0071), while school districts receiving reduced meals did not (*p* = 0.5544). When students receiving free meals and reduced meals were evaluated together, there was no association with increased rates of full immunization (*p* = 0.135). Religious exemptions were used consistently more for students that received free meals, reduced meals, and free and reduced meals together (*p* = 2.5 × 10^−12^, 2.7 × 10^−15^ and 1.3 × 10^−24^ respectively). Personal exemptions displayed decreased rates on students receiving free meals (*p* = 3.8 × 10^−14^) and free and reduced meals together (*p* = 1.9 × 10^−9^). Medical exceptions were also more frequent in school districts with a higher percentage of students receiving free meals (*p* = 0.0199) but less frequent in districts with higher percentage of students receiving reduced meals (*p* = 0.0274). In Table 2, we noted significant associations flipping in direction between free meals and reduced meals (medical and personal exemptions). We speculate that this observation may derive from differences in the context of poverty between locations. Records in process we less often observed in districts with reduced meals (*p* = 0.0048). Incomplete, no-records and overall compliance displayed no associations to free or reduced meal proportions.

Subsequent modeling of race indicators on vaccination status are presented in Table 3. Although districts with larger White or larger Hispanic populations are both a predominant majority in Colorado, their vaccination rates showed some key differences that were not evident from factor analysis. The modeling analysis showed that school districts with larger Hispanic populations were more likely to be fully immunized (*p* = 5.5 × 10^−15^), while districts with larger White populations were predictive for lower vaccination rates (*p* = 9.9 × 10^−10^). The trends for exemptions were overall reversed for these two populations (based on estimate effect directions). Districts with larger White populations tended to have a higher rate of exemptions in general while districts with larger Hispanic populations had lower rates. Districts with white populations tended to have more personal (*p* = 8.3 × 10^−13^) and religious exemptions (*p* = 0.0240). Other racial groups included in the analysis showed significant patterns, such as Native Americans claiming more personal exemptions (*p* = 1.6 × 10^−11^) and Pacific Islanders being less compliant overall (*p* = 1.4 × 10^−16^). A combined modeling evaluation that included the interaction of racial and poverty indicators revealed many significant associations only visible with a sub-stratified analysis (Table 4). This stratified modeling analysis revealed that some synergistic associations exist (interaction effect: race indicator x poverty indicator) and suggests that racial and cultural disparities are often more complicated to explain.

Overall, the associations found using the modeling approach confirm what was observed in the factor analysis suggesting that the type of exceptions used by parents and the completeness of the records is correlated to economic and racial differences in the population. These differences are likely influenced by cultural and counterintuitive socioeconomic differences.

## 4. Discussion

The focus of this study was to explore the associations between immunization status within school districts and different demographic indicators such as race and poverty level in Colorado. Our findings do not indicate causality but highlight that sociodemographic and cultural characteristics are antecedents to vaccine hesitancy behaviors (noting again that only race, ethnicity and poverty were evaluated in this study). In the state of Colorado under the Code of Regulations (6 CCR 1009-2 effective on 14 August 2018), medical exemption requires the signature of a health professional indicating the exemption while religious and personal exceptions only require the parents’ approval. There is a large amount of ambiguity in and very little oversight in the validity of these exceptions, however, previous reports suggest that these exemptions have remained low enough to not represent a threat to herd immunity [24]. Our data suggest exemptions are differentially used according primarily to racial background even where the requirement is same for some of them (both religious and personal exemptions only require parent’s approval). The largest difference in the type of exemptions used by parents was observed to occur predominantly between racial background; the exception type choice of predominantly White and predominantly Hispanic districts sit at opposite ends of the spectrum. A pattern observed in this study was that districts with larger proportions of students receiving free and reduced lunch were more likely to be fully immunized and used fewer personal than medical and religious exemptions. This is similar to findings in COVID-19 vaccination hesitancy [20]. Complex interactions between religion, socioeconomic status, and parental psychology likely contribute to this relationship. Children with lower socioeconomic status may be more religious [25], resulting in an increase in religious exemptions. Furthermore, Colorado is one of 15 states that allows for both religious and personal exemptions to childhood vaccinations [26,27]. States that have laws that allow for increased exemptions, such as Colorado, have lower rates of childhood vaccination [27,28]. Overall, allowing more exemption types (medical and nonmedical) has shown to be associated with higher vaccination hesitancy [29,30], Colorado being no exception.

Socioeconomic status has been shown to influence vaccination rates, but the effect has been mixed and has mostly been explored in the United States. Some studies have shown that lower socioeconomic status and education (indicated by free and reduced lunch in this study) is associated with increased vaccine hesitancy [31,32,33,34]. However, other studies have shown that higher socioeconomic status and increased level of parental education are associated with increased vaccine hesitancy behaviors as well [34,35,36,37,38,39]. The data analysis in this study supported the latter studies with children with increased socioeconomic status being associated with higher vaccination hesitancy [20,40,41].

Perceptions among different racial and ethnic populations are an important factor influencing vaccination hesitancy. Hispanics have been shown to have increased confidence in childhood and influenza vaccines, albeit increased risk perception, and are more likely to vaccinate their children compared to non-Hispanics [34,42]. This finding is similar with COVID-19 vaccinations [2,43]. In this study, even when districts with larger Hispanic and poverty populations overlapped, they still displayed a higher association with being fully immunized. Social, cultural, and parental psychology of Hispanics are all likely contributing to increased vaccination rates among the Hispanics despite poverty. In this study, districts with predominantly White populations displayed higher vaccination hesitancy. Some studies have shown that increased socioeconomic privilege is associated with increased vaccine hesitancy [44]. A study performed in California from 2007 to 2013 showed that higher income, greater White population, and private school attendance correlated with increased rates of personal and religious exemptions to childhood vaccination [45]. Attitudes about childhood vaccinations have been evaluated and shown that a significant number of parents make vaccination decisions based on sources other than the pediatrician (62%) or mass media (12%) and this trend is evident even when adjusting for education and nationality [46]. Since this narrative involves cultural and racial aspects, it is important to note how these findings may be specifically relevant only within defined regions in the United States.

### Limitations

Limitations in this study are associated with the aggregate nature of the data where vaccination status, demographics, and free and reduced meals percentages were obtained on a per school district basis and the characteristics of these districts are not homogenous, therefore there will always be an amount of inherent bias in these types of analysis that may affect any inference and thus cannot be used to link directly to causality. In this study, racial and ethnic denominations are self-identified as defined by Census guidelines, people are allowed to select more than one race or ethnicity [47]. The study did not look at individual motivations behind parental decisions to vaccinate or not vaccinate. The overarching rationale of the narrative presented in the paper is Colorado specific and is limited by the state’s characteristics. Colorado has a sizeable White and Hispanic population, but has a much lower representation of Black, Multiracial, Pacific Islander, Asian, or Native American populations which may hide peculiarities of those populations. Replicating this study in other states with different characteristics may clarify the context of the findings presented and could even be explored individually within specific vaccinations. It would be interesting to know how other countries pair on these trends. Based on the narrative presented, future efforts should be directed to the characterization of environmental [41,45] cultural differences [34] of vaccination attitudes and by clearly identifying stakeholders [48], which can improve effective vaccination campaign efforts that do not alienate the unvaccinated or strengthen their resistance [49]. Dealing with low vaccination rates among privileged populations is a challenge, since their privilege allows them to decide against vaccination. Continued public education is requited to emphasize the public health benefit of vaccinations. Mandatory childhood vaccinations are a potential solution to this issue but have consistently been followed with some backlash [50]. An example of a proposed mandatory vaccination campaign that met significant resistance is the HPV vaccinations [51]. Even if these childhood vaccination mandates are implemented, the public in general is generally not supportive of sanctioning people refusing vaccination [52,53,54] thus enforcement of these policies is likely unattainable.

## 5. Conclusions

Colorado is a highly educated predominantly White state with high median household incomes, yet still struggles with vaccination for childhood diseases. Childhood vaccination hesitancy is not a problem exclusively affecting the underprivileged. Many variables affect parental vaccination decisions for their children, and these variables are predominantly influenced by cultural and socioeconomic factors. These factors do not necessarily follow the traditional narratives of higher levels of poverty or minority status being the culprits of hesitancy. Even though these factors may not be causative, they should be always considered when developing vaccination campaigns and influence vaccination requirements that are relevant to specific populations. It may be necessary to develop completely new educational approaches to target privileged populations.

## Figures and Tables

**Figure 1 vaccines-10-01579-f001:**
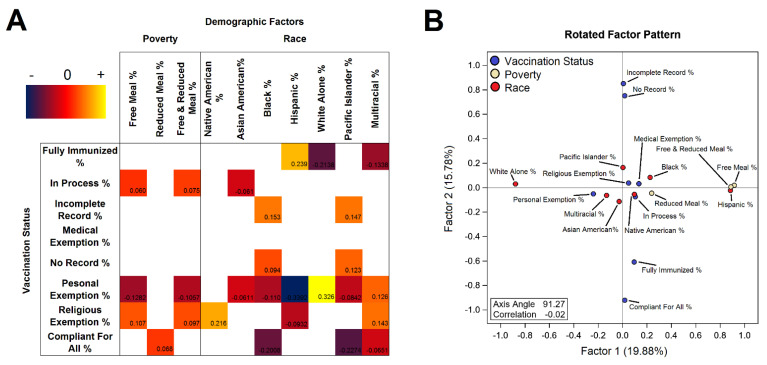
Correlation and factor analysis of all vaccination hesitancy and demographic factors in Colorado. (**A**) Pearson correlations (**B**) Oblimin rotated factor pattern for all vaccination compliance.

**Table 1 vaccines-10-01579-t001:** Colorado school district demographic summarized data. Mean, SD, Minimum and Maximum values estimated from School District aggregate data.

	Variable	Mean	Std Dev	Minimum	Maximum
	School Population	4411.74	11,575.23	12	85,202
	Number of Facilities	9.28	22.26	1	198
Poverty	Free Meal %	36.7%	16.6%	1.6%	91.2%
	Reduced Meal %	9.8%	4.7%	0.9%	31.6%
	Free & Reduced Meal %	46.3%	18.5%	2.1%	96.6%
Race	Native American %	1.0%	3.0%	0.0%	31.9%
	Asian %	0.9%	1.3%	0.0%	9.0%
	Black %	1.5%	2.7%	0.0%	22.1%
	Hispanic %	29.1%	21.0%	0.0%	95.7%
	White Alone %	64.6%	21.7%	3.3%	100.0%
	Pacific Islander %	0.2%	0.3%	0.0%	2.9%
	Multiracial %	2.7%	2.3%	0.0%	15.2%

**Table 2 vaccines-10-01579-t002:** Free meal and reduced meal association to vaccination status or exemption type. Asterisk (*) indicates significance at *p* ≤ 0.05. Double asterisk (**) indicated significance at Bonferroni adjusted P (adjusted to 80 tests).

Variable		Fully Immunized %	In Process %	Incomplete Record %	Medical Exemption %	No Record %	Personal Exemption %	Religious Exemption %	Compliant For All %
Free Meal %	Estimate	0.035	0.003	0.007	0.002	−0.001	−0.056	0.010	0.018
	Std Error	0.013	0.004	0.008	0.001	0.004	0.007	0.001	0.014
	*p*-Value	0.0071 *	0.4494	0.3767	0.0199 *	0.8350	3.8 × 10^−14^ **	2.5 × 10^−12^ **	0.1936
Reduced Meal %	Estimate	−0.016	−0.030	0.020	−0.005	0.012	0.004	0.026	−0.021
	Std Error	0.026	0.011	0.018	0.002	0.010	0.015	0.003	0.027
	*p*-Value	0.5544	0.0048 *	0.2473	0.0274 *	0.2371	0.8135	2.7 × 10^−15^ **	0.4407
Free & Reduced Meal %	Estimate	0.017	0.000	0.008	0.001	0.002	−0.039	0.013	0.006
	Std Error	0.011	0.004	0.007	0.001	0.005	0.006	0.001	0.012
	*p*-Value	0.1350	0.9794	0.2331	0.1780	0.6074	1.9 × 10^−9^ **	1.3 × 10^−24^ **	0.5950

**Table 3 vaccines-10-01579-t003:** Race association to vaccination status or exemption type.

Variable		Fully Immunized %	In Process %	Incomplete Record %	Medical Exemption %	No Record %	Personal Exemption %	Religious Exemption %	Compliant For All %
Native American %	Estimate	−0.306	−0.006	−0.090	−0.006	−0.008	0.438	0.033	0.276
	Std Error	0.108	0.025	0.053	0.005	0.035	0.065	0.011	0.120
	*p*-Value	0.0047 *	0.8246	0.0940	0.2438	0.8246	1.6 × 10^−11^ **	0.0018 *	0.0214 *
Asian %	Estimate	−0.264	−0.038	0.221	0.020	−0.012	0.045	0.067	0.034
	Std Error	0.216	0.066	0.122	0.013	0.083	0.123	0.024	0.223
	*p*-Value	0.2208	0.5690	0.0702	0.1382	0.8848	0.7135	0.0053 *	0.8794
Black %	Estimate	0.068	−0.035	−0.037	0.003	0.007	0.110	0.037	0.553
	Std Error	0.081	0.027	0.048	0.005	0.033	0.046	0.009	0.081
	*p*-Value	0.4040	0.1949	0.4454	0.6167	0.8314	0.0163 *	8.1 × 10^−5^ **	1.3 × 10^−11^ **
Hispanic %	Estimate	0.120	−0.006	0.004	0.001	−0.007	−0.094	−0.005	−0.002
	Std Error	0.015	0.004	0.008	0.001	0.005	0.009	0.002	0.018
	*p*-Value	5.5 × 10^−15^ **	0.0942	0.5765	0.2292	0.1368	1.7 × 10^−26^ **	0.0019 *	0.9152
White Alone %	Estimate	−0.094	0.005	−0.002	−0.001	0.008	0.063	0.004	−0.046
	Std Error	0.015	0.004	0.008	0.001	0.005	0.009	0.002	0.018
	*p*-Value	9.9 × 10^−10^ **	0.1885	0.8022	0.1330	0.0941	8.3 × 10^−13^ **	0.0240 *	0.0105 *
Pacific Islander %	Estimate	0.646	−0.361	0.782	0.062	0.354	−1.126	−0.008	−3.916
	Std Error	0.485	0.185	0.301	0.038	0.212	0.269	0.057	0.472
	*p*-Value	0.1826	0.0513	0.0093 *	0.0998	0.0953	2.9 × 10^−5^ **	0.8868	1.4 × 10^−16^ **
Multiracial %	Estimate	−0.671	0.149	−0.044	0.015	−0.054	0.595	−0.028	0.492
	Std Error	0.085	0.029	0.051	0.006	0.035	0.047	0.010	0.086
	*p*-Value	3.5 × 10^−15^ **	4.0 × 10^−7^ **	0.3869	0.0120 *	0.1263	1.8 × 10^−35^ **	0.0048 *	1.0 × 10^−8^ **

Asterisk (*) indicates significance at *p* ≤ 0.05. Double asterisk (**) indicated significance at Bonferroni adjusted P (adjusted to 80 tests).

**Table 4 vaccines-10-01579-t004:** Free and reduced meal by race interaction association to vaccination status or exemption type.

Effect	Fully Immunized %	In Process %	Incomplete Record %	Medical Exemption %	No Record %	Personal Exemption %	Religious Exemption %	Compliant For All %
Native American %	3.5 × 10^−8^ **	0.1445	0.7556	0.5826	0.4432	1.1 × 10^−14^ **	0.0521	0.3003
Free & Reduced meal %	0.6856	0.5669	0.2309	0.2425	0.4199	0.0004 *	2.7 × 10^−23^ **	0.5418
Native American % by Free & Reduced meal % interaction	9.8 × 10^−7^ **	0.1259	0.8606	0.7483	0.3855	1.4 × 10^−8^ **	0.2421	0.9569
Asian %	0.9915	0.6608	0.0271 *	0.0996	0.4928	0.1177	0.7184	0.0034 *
Free & Reduced meal %	0.1854	0.8223	0.0435 *	0.7448	0.9441	1.8 × 10^−9^ **	2.3 × 10^−18^ **	0.0733
Asian % by Free & Reduced meal % interaction	0.9282	0.8151	0.1355	0.0039 *	0.5402	0.2672	0.2632	0.0097 *
Black %	0.9052	0.3134	0.2235	0.2033	0.1359	0.0366 *	0.7356	0.3307
Free & Reduced meal %	0.2924	0.8403	0.5233	0.7426	0.2199	6.6 × 10^−7^ **	9.3 × 10^−17^ **	0.0993
Black % by Free & Reduced meal % interaction	0.6932	0.5453	0.3374	0.1508	0.1208	0.2678	0.4144	0.0004 *
Hispanic %	4.1 × 10^−9^ **	0.7442	0.8217	0.0902	0.3619	5.0 × 10^−23^ **	0.0262 *	0.0307 *
Free & Reduced meal %	0.2488	0.2759	0.3074	0.3917	0.5440	2.3 × 10^−11^ **	5.5 × 10^−37^ **	0.0289 *
Hispanic % by Free & Reduced meal % interaction	0.0245 *	0.6258	0.7633	0.0393 *	0.7873	1.6 × 10^−8^ **	1.0 × 10^−9^ **	0.0116 *
White %	0.0001 **	0.6193	0.7683	0.1310	0.4881	2.4 × 10^−9^ **	0.0052 *	1.6 × 10^−4^ **
Free & Reduced meal %	0.3684	0.8777	0.9300	0.0349 *	0.7420	0.0888	0.0319 *	0.0103 *
White % by Free & Reduced meal % interaction	0.3542	0.8966	0.5963	0.0459*	0.9440	0.0009 *	6.7 × 10^−10^ **	0.0064 *
Pacific Islander %	0.0932	0.8695	0.0027 *	0.1028	0.8454	3.9 × 10^−8^ **	0.1447	1.7 × 10^−7^ **
Free & Reduced meal %	0.0872	0.8400	0.0882	0.7239	0.8414	1.1 × 10^−11^ **	1.1 × 10^−22^ **	0.7747
Pacific Islander % by Free & Reduced meal % interaction	0.2664	0.4136	0.0452 *	0.0079 *	0.2685	1.4 × 10^−4^ **	0.1007	0.2362
Multiracial %	6.7 × 10^−8^ **	0.0068 *	0.2398	0.0174 *	0.3835	4.7 × 10^−25^ **	0.0870	0.1799
Free & Reduced meal %	0.3619	0.0014 *	0.0610	0.0267 *	0.1948	0.6899	1.9 × 10^−11^ **	0.7098
Multiracial % by Free & Reduced meal % interaction	0.0662	6.4 × 10^−8^ **	0.1055	0.1724	0.1017	4.2 × 10^−7^ **	0.0682	0.3395

Asterisk (*) indicates significance at *p* ≤ 0.05. Double asterisk (**) indicated significance at Bonferroni adjusted P (adjusted to 168 tests).

## Data Availability

Data used in the study is available publicly from the Colorado Department of Education and the Colorado Department of Public Health the Environment. Curated datasets can be made available with a reasonable request to the authors.

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
