# Peer review of "Preventable Disease, the Case of Colorado: School District Demographics and Childhood Immunizations"

_vaccines, 2022, doi:10.3390/vaccines10101579_

Round 1
Reviewer 1 Report
The topic of this paper is important. Given that vaccination rates are not ideal understanding what variables affect that will enable better targeting of attempts to increase vaccination rates.
My review is qualified because I have limited understanding of the statistics involved. It may be that if the audience is predominantly those with statistical knowledge that this is satisfactory but I am unable to judge this. I am a clinician with significant experience of giving vaccines, overseeing vaccination programmes and reflecting on how to over come vaccine hesitancy.
I am unable to evaluate the claims made in the discussion, although if valid they are certainly interesting.
Nowhere is simple data given on vaccination rates for any variable...the only data presented is on correlation between variables. What is the vaccination rate for each ethnic group? What is the vaccination rate for each of the "free lunch" categories?
It would be helpful to provide information on how ethnic classification is arrived at. Is this self identified? What guidance is given on prioritising race (if a person identifies as Hispanic but has a "white" grandfather are they counted as multiracial or Hispanic?) Is there information on the accuracy of ethnicity classification?
It would similarly be helpful to understand the criterion for free lunches; how poor are these families?
Line 37 "Childrens" should be Children
Line 64 This not his evaluate "and" identify trends
Author Response
Reviewer 1
My review is qualified because I have limited understanding of the statistics involved. It may be that if the audience is predominantly those with statistical knowledge that this is satisfactory but I am unable to judge this. I am a clinician with significant experience of giving vaccines, overseeing vaccination programmes and reflecting on how to over come vaccine hesitancy.
Thank you for your review, we present this study with a target audience that includes, public health experts, analysts, clinicians and public in general. We appreciate your comments and suggestions as they help us polish our report from your perspective.
I am unable to evaluate the claims made in the discussion, although if valid they are certainly interesting.
We hope to have added enough of the additional information suggested by the reviewers to allow him to evaluate accordingly our claims.
Nowhere is simple data given on vaccination rates for any variable...the only data presented is on correlation between variables.
This data was indeed supplied as Supplemental Table 1. This data mentioned in the text and is noted at the Supplemental Materials section. We did not see appropriate to report in our main text these values as they are not gathered by us and are reported and available from the Colorado Department of Public Health and Environment. Also, as noted in our data availability statement, we can provide curated dates if requested to the authors.
What is the vaccination rate for each ethnic group? What is the vaccination rate for each of the "free lunch" categories?
This is a great suggestion; however, we are not able to provide it. This study was performed on state-wide district level aggregate data collected from two different agencies, Colorado Department of Education and Colorado Department of Public Health and Environment. Vaccination rates are provided by the Department of Public Health and Environment while racial, ethnic and free and reduced meal data are provided by the Department of Education. Unfortunately, the data is provided without the ability of cross matching. That is the exact reason we used correlation, dimensional analysis and models to assess indirectly the trends we observed.
In an unofficial argument, the reason for that information not being available has to do with privacy and political motivations. In the United States, race is a very delicate topic due to the large diversity and inherent problems with disparities that are often associated to political ideologies. Poverty is not different. Other countries with more homogenous racial and ethnic representation may have more uniform opinion that facilitate these discussions. For that reason, our study is special and important in that it attempts to open up the discussion by pointing out these differences and their impact in our society.
It would be helpful to provide information on how ethnic classification is arrived at. Is this self identified? What guidance is given on prioritising race (if a person identifies as Hispanic but has a "white" grandfather are they counted as multiracial or Hispanic?) Is there information on the accuracy of ethnicity classification?
We added this passage and added a citation as clarification in our limitations section: “In this study, racial and ethnic denominations are self-identified as defined by Census guidelines, people are allowed to select more than one race or ethnicity.”
For additional context for the reviewer, author I. Zapata defines himself as Hispanic-White therefore for purposes of this study he would be placed along with the “Hispanic” population but not with the “White, non-Hispanic”. In general, these classifications in the United States are very standardized.
It would similarly be helpful to understand the criterion for free lunches; how poor are these families?
We have added this information and a citation into the methods section that reads: “This eligibility is determined by guidelines set and adjusted every year that take into ac-count the household size and income. In 2022, a married couple with two children will require reporting less than $36,075 or less than $51,338 to qualify for free or reduced meals, respectively.”
Line 37 "Childrens" should be Children
Corrected, thank you.
Line 64 This not his evaluate "and" identify trends
And was added, thank you.
Reviewer 2 Report
Review of Preventable Disease, the Case of Colorado: School District Demographics and Childhood Immunizations
This manuscript discusses findings from aggregate data on the vaccination status of children in public schools in Colorado. It addresses the correlation of and interaction between different factors (race, ethnicity, and free / reduced meal eligibility as a proxy for poverty) and vaccination status and exemption. It is important to be able to tease out different patterns at a population level, and vaccine hesitancy is highly relevant today. The paper is well-written and easy to read.
That said, I have outlined several ways to improve the paper, both major and minor.
Major:
In the introduction, you start with a WHO definition of vaccine hesitancy that is behavioral and then describe vaccine hesitancy as both attitudinal and behavioral. I do not think the WHO definition adds much to the paper. There are many papers that offer more appropriate definitions of vaccine hesitancy for your context.
The data that you have, as you note, is correlational rather than causal. Thus, you must precede with caution in terms of your inferences. In several places, you imply that the demographics caused the vaccine hesitancy. For example, line 167: “Increased White population in the district resulted in more personal and religious exemptions” – I think is correlated with is more appropriate. Another example is line 183: “… is strongly influenced by economic and racial differences in the population and these are likely very intimate influenced by cultural and counterintuitive socioeconomic differences.” This may be true, but it is outside of the scope of your data. More references (especially from this journal) and less forceful assertions would strengthen your paper. You could elaborate on this carefully in the discussion, building on findings from other studies, rather than in the results section.
Other aspects of the discussion are not clear. This sentence in particular (lines 189-191): “The factors influencing vaccine hesitancy were likely derived from sociodemographic and cultural differences; this data from the state of Colorado provides a great example of that.” In your study, you have sociodemographic differences, but it is not clear what “factors” you mean here. You seem to imply that the variables in your model represent something else. I would focus on what you do have, which is evidence of interesting correlations and interaction effects, rather than other factors influencing vaccine hesitancy.
Many of your findings confirm previous studies, which should be cited (correlation between affluence / White race and vaccine hesitancy is documented, especially in certain areas of the US), see:
Aw, J., Seng, J. J. B., Seah, S. S. Y., & Low, L. L. (2021). COVID-19 vaccine hesitancy—A scoping review of literature in high-income countries. Vaccines, 9(8), 900.
In the findings, you note important differences between populations that have more free meals versus reduced meals. This deserves more discussion. What is the difference between income levels for families getting free vs. reduced price meals? Why do you think this is significant? I am surprised that this change causes some of the correlations to flip.
Furthermore, if you recommend different vaccine campaigns for different groups (segments), you will need to include more literature about messaging and existing campaigns for vaccine acceptance.
Finally, I would like more background on why you chose your analytical approach. Could additional models incorporate more relationships?
Overall, I believe that you could add more references from this journal that are relevant to vaccine hesitancy.
Minor:
· Be clear that individual compliance parameters are the different levels of vaccine adoption – sometimes it is clear and sometimes not
· Typo on line 64 missing first letter “T”
· Another typo on line 64: “evaluate identify” – pick one or the other
· Line 69-70: be consistent with the name of the health department
· Line 93: missing word or punctuation between “(GLMMs)” and “the total…”
· Line 158: I would suggest starting a new paragraph at “Subsequent…” to improve readability
· Lines 182-183 “type of completeness and type of exceptions used by parents” implies that type of completeness was used by parents
· Line 235: “significant amount of parents” – should be “significant number of parents”
· Lines 247-250: you note yourselves that the % of non-White and non-Hispanic residents limits potential findings for those groups but previously you still discussed findings for these groups. Although statistically significant, the numbers are quite low as you mention.
· Line 258-259: In your limitations you note that Colorado is different from other areas of the US, but then here say that is provides a “great context that is applicable elsewhere.”
Thank you for the opportunity to review and good luck with your research.
Author Response
Reviewer 2
This manuscript discusses findings from aggregate data on the vaccination status of children in public schools in Colorado. It addresses the correlation of and interaction between different factors (race, ethnicity, and free / reduced meal eligibility as a proxy for poverty) and vaccination status and exemption. It is important to be able to tease out different patterns at a population level, and vaccine hesitancy is highly relevant today. The paper is well-written and easy to read.
That said, I have outlined several ways to improve the paper, both major and minor.
We appreciate the effort made by the reviewer to review our paper. We hope to present this study on vaccination hesitancy with the goal of opening up the discussion of these delicate topics of race, ethnicity and poverty. All comments were thoroughly considered and we hope to have addressed all your concerns.
Major:
In the introduction, you start with a WHO definition of vaccine hesitancy that is behavioral and then describe vaccine hesitancy as both attitudinal and behavioral. I do not think the WHO definition adds much to the paper. There are many papers that offer more appropriate definitions of vaccine hesitancy for your context.
We used that definition as it is one used very often in the many papers we reviewed. We are unsure of what the reviewer considers to be less appropriate of this definition. We made no changes for this comment.
The data that you have, as you note, is correlational rather than causal. Thus, you must precede with caution in terms of your inferences. In several places, you imply that the demographics caused the vaccine hesitancy. For example, line 167: “Increased White population in the district resulted in more personal and religious exemptions” – I think is correlated with is more appropriate. Another example is line 183: “… is strongly influenced by economic and racial differences in the population and these are likely very intimate influenced by cultural and counterintuitive socioeconomic differences.” This may be true, but it is outside of the scope of your data. More references (especially from this journal) and less forceful assertions would strengthen your paper. You could elaborate on this carefully in the discussion, building on findings from other studies, rather than in the results section.
This is a fabulous observation, we agree completely with the reviewer. Therefore, we added several warnings throughout the paper (results, discussion, and the conclusion) indicating that causality should not be inferred from this study.
We understand the issue. We have softened the wording of some of our statements. Indeed, these statements may be too strong that deviate the attention to something we did not target directly in our study. We hope our toned down version is more appropriate.
Other aspects of the discussion are not clear. This sentence in particular (lines 189-191): “The factors influencing vaccine hesitancy were likely derived from sociodemographic and cultural differences; this data from the state of Colorado provides a great example of that.” In your study, you have sociodemographic differences, but it is not clear what “factors” you mean here. You seem to imply that the variables in your model represent something else. I would focus on what you do have, which is evidence of interesting correlations and interaction effects, rather than other factors influencing vaccine hesitancy.
We agree with the reviewer. This argument was ambiguous and does not connect well with the rest of the idea. This was rewritten and now reads as: “Our findings do not indicate causality but highlight that sociodemographic and cultural characteristics are associated with vaccine hesitancy (noting again that only race, ethnicity and poverty were evaluated in this study).”
Many of your findings confirm previous studies, which should be cited (correlation between affluence / White race and vaccine hesitancy is documented, especially in certain areas of the US), see:
Aw, J., Seng, J. J. B., Seah, S. S. Y., & Low, L. L. (2021). COVID-19 vaccine hesitancy—A scoping review of literature in high-income countries. Vaccines, 9(8), 900.
We were aware of this paper. We debated about including it because of a previous reviews comment we got elsewhere where our argument was being put heavily (and incorrectly) to the COVID perspective, we got rejected elsewhere because our argument was seen primarily as COVID derived. In our paper, we present a case that precedes COVID. We decided to limit our arguments on COVID because of this concern. However, we agree that this is an important citation and we have now reintroduced it in several arguments. We also added more references to provide weight to this argument.
In the findings, you note important differences between populations that have more free meals versus reduced meals. This deserves more discussion. What is the difference between income levels for families getting free vs. reduced price meals?
These thresholds are now described in the methods section; this was also brought up by reviewer 1. We added the following text with an additional citation for the source: “This eligibility is determined by guidelines set and adjusted every year that take into ac-count the household size and income. In 2022, a married couple with two children will require reporting less than $36,075 or less than $51,338 to qualify for free or reduced meals, respectively”.
Why do you think this is significant? I am surprised that this change causes some of the correlations to flip.
Since the reviewer is talking about correlations, we assume he is talking about Figure 1A. There indeed, we observe a flip in the directions of the correlations to specific exemption types (personal vs religious). This suggests that the selection of these two specific types of exemptions is opposite. Table 2 also displays interesting patterns where medical and personal exceptions have opposite estimates associated to free meal and reduced meal. This observation is very interesting. We speculate that this is related to specific income structures associated to specific school districts where poverty in one place has different connotations than in other districts.
Said in different words, the threshold that defines eligibility to free meals and reduced meals (an indicator of poverty) does not imply the same context which may involve differences in living costs across regions in the state. This may be also associated indirectly to race. It is not the same to live on less than $36,075 in metropolitan Denver than living on the same income in Trinidad (a mostly rural town at the south border of the state).
We added this text to the results: “In table 2, we noted significant associations flipping in direction between free meals and reduced meals (medical and personal exemptions). We speculate that this observation may derive from differences in the context of poverty between locations.”
Furthermore, if you recommend different vaccine campaigns for different groups (segments), you will need to include more literature about messaging and existing campaigns for vaccine acceptance.
This is a good point; however, we do not know what the answer would be. How can you force the privileged to do something they do not want without causing backlash? We added a brief discussion about this situation at the end of our discussion. There we highlight the problem and some of the issues associated with it. This section reads: “Dealing with low vaccination rates among privileged populations is a challenge, since their privilege allows them to decide against vaccination. Continued public education is requited to emphasize the public health benefit of vaccinations. Mandatory childhood vaccinations are a potential solution to this issue but have consistently been followed with some backlash. An example of a proposed mandatory vaccination campaign that met significant resistance is the HPV vaccinations. Even if these childhood vaccination mandates are implemented, the public in general is generally not supportive of sanctioning people refusing vaccination thus enforcement of these policies is likely unattainable..”
Finally, I would like more background on why you chose your analytical approach. Could additional models incorporate more relationships?
This is an important question. We are aware that our methodology is unconventional in the public health field. This methodology was selected because of the characteristics of the data available. The data we used in this study was provided by two separate government agencies: the Colorado Department of Public Health and Environment and the Colorado Department of Education. Each agency provides data as aggregate per district where it is not possible to sub-stratify per race, ethnicity and poverty. Our methods were chosen to detect indirectly those associations. Both our dimensional analysis and modeling approach provide alternative methods to reach similar answers.
Unfortunately, although data should be available to the public, it may not be. In America data disclosure to government agencies is not even across regions and it is often incomplete because of privacy and political motives.
Overall, I believe that you could add more references from this journal that are relevant to vaccine hesitancy.
We have expanded our citations as suggested. We have added a total of 12 more citations with more than half of them from this journal.
Minor:
Be clear that individual compliance parameters are the different levels of vaccine adoption – sometimes it is clear and sometimes not
We made the correction in the methods.
Typo on line 64 missing first letter “T”
This was corrected, thank you.
Another typo on line 64: “evaluate identify” – pick one or the other
This was noted as well by reviewer 1. We corrected it to “evaluate and identify”.
Line 69-70: be consistent with the name of the health department
Thank you for noticing that, we corrected all instances.
Line 93: missing word or punctuation between “(GLMMs)” and “the total…”
This was corrected, thank you.
Line 158: I would suggest starting a new paragraph at “Subsequent…” to improve readability
We have made the change.
Lines 182-183 “type of completeness and type of exceptions used by parents” implies that type of completeness was used by parents
We see the ambiguity in the argument, therefore we changed the sentence to: “…analysis suggesting that the type of exceptions used by parents and the completeness of the records is correlated to economic…”
Line 235: “significant amount of parents” – should be “significant number of parents”
Thank you, we made the change.
Lines 247-250: you note yourselves that the % of non-White and non-Hispanic residents limits potential findings for those groups but previously you still discussed findings for these groups. Although statistically significant, the numbers are quite low as you mention.
According to US Census data, Colorado is a predominant White state followed by Hispanics, which could also be White, although evaluated separately according to census denominations. Proportions for other groups are low, but accurate. Their low representation is not due to limited sampling, they are just not represented. For this reason, we decided to include all groups; however we made an effort on not making strong inference based on numeric differences because of this limitation.
Line 258-259: In your limitations you note that Colorado is different from other areas of the US, but then here say that is provides a “great context that is applicable elsewhere.”
In general, Colorado is not even average performer in terms of socioeconomic factors among all states in the Nation. Colorado is a top performer; however, even with this advantage there are some particular factors where it counterintuitively underperforms such as having a high vaccination hesitancy rate. Part of the narrative in our paper is to highlight how vaccination hesitancy cannot be classified as a problem of the poor or the minorities. It is easy to blame the underprivileged as an ubiquitous factor associated to poor outcomes. In other states or regions with much poorer performance, it may be harder to evidence our argument.
We have reworded our argument as follows: “Colorado is a highly educated predominantly White state with high median house-hold incomes, yet still struggles with vaccination for childhood diseases. Childhood vaccination hesitancy is not a problem exclusively affecting the underprivileged.”
Thank you for the opportunity to review and good luck with your research.
Round 2
Reviewer 1 Report
Thankyou I am happy to support publication of this paper
Author Response
Thanks to you for helping us improve our work.
Reviewer 2 Report
Thank you for your thoughtful responses to my previous comments. You have addressed most of my comments thoroughly.
I have two remaining concerns.
The first is about how you define vaccine hesitancy, as I noted previously. I understand that you want to keep the WHO definition. I would then be clear that you believe hesitancy is a behavior only – as defined by WHO – and that the attitudes and beliefs and socioeconomic factors you discuss are antecedents to this behavior (delay or refusal). For example, on line 44 you could say that the causes (or antecedents) of vaccine hesitancy are multifactorial. The behavior itself is not multifactorial – it’s actually quite simple: no to vaccines. I think it would add clarity to be consistent throughout the manuscript about what you mean exactly by hesitancy.
The second concern is around presenting vaccine hesitancy among wealthier groups as completely unexpected. I agree that many talk about vaccine hesitancy as being limited to certain disadvantaged groups, but research already exists that suggests in fact one segment of the vaccine hesitant is in face quite well-off and educated and primarily white (Roberts et al. 2022, Reich 2014). I would suggest adding these references so that your claim “Childhood vaccination hesitancy is not a problem exclusively affecting the underprivileged” has a bit more context.
Minor:
It appears that there are paragraph breaks on Line 187 and 195 that do not belong there.
References 45 and 47 appear to have typos.
Overall, I enjoyed reading your paper again and wish you the best of luck with your research.
Citations:
Reich, J. A. (2014). Neoliberal mothering and vaccine refusal: imagined gated communities and the privilege of choice. Gender & Society, 28(5), 679-704.
Roberts, H. A., Clark, D. A., Kalina, C., Sherman, C., Brislin, S., Heitzeg, M. M., & Hicks, B. M. (2022). To vax or not to vax: Predictors of anti-vax attitudes and COVID-19 vaccine hesitancy prior to widespread vaccine availability. Plos one, 17(2), e0264019.
Author Response
Thank you for your thoughtful responses to my previous comments. You have addressed most of my comments thoroughly.
I have two remaining concerns.
The first is about how you define vaccine hesitancy, as I noted previously. I understand that you want to keep the WHO definition. I would then be clear that you believe hesitancy is a behavior only – as defined by WHO – and that the attitudes and beliefs and socioeconomic factors you discuss are antecedents to this behavior (delay or refusal). For example, on line 44 you could say that the causes (or antecedents) of vaccine hesitancy are multifactorial. The behavior itself is not multifactorial – it’s actually quite simple: no to vaccines. I think it would add clarity to be consistent throughout the manuscript about what you mean exactly by hesitancy.
We understand the concern and we realize this was what you meant from the first review. We appreciate the suggestion. We have made the corrections. That argument reads now as: “Causes or antecedents to vaccine hesitancy are multifactorial and hinge on parental attitudes and beliefs; however, they lead to a simple behavior of not accepting vaccines.”
The second concern is around presenting vaccine hesitancy among wealthier groups as completely unexpected. I agree that many talk about vaccine hesitancy as being limited to certain disadvantaged groups, but research already exists that suggests in fact one segment of the vaccine hesitant is in face quite well-off and educated and primarily white (Roberts et al. 2022, Reich 2014). I would suggest adding these references so that your claim “Childhood vaccination hesitancy is not a problem exclusively affecting the underprivileged” has a bit more context.
These are excellent recommendations of two very strong papers. We have added these references.
Minor:
It appears that there are paragraph breaks on Line 187 and 195 that do not belong there.
We see the issue, it was the “-“ in the “E-10” we corrected it by adding an extra space that shifter the whole number to the next lane, we hope this error does not show up again in the proof, as they like to remove double spaces.
References 45 and 47 appear to have typos.
These now are corrected
Overall, I enjoyed reading your paper again and wish you the best of luck with your research.
We thank you for all your comments and suggestions. You have helped us improve our paper.
Citations:
Reich, J. A. (2014). Neoliberal mothering and vaccine refusal: imagined gated communities and the privilege of choice. Gender & Society, 28(5), 679-704.
Roberts, H. A., Clark, D. A., Kalina, C., Sherman, C., Brislin, S., Heitzeg, M. M., & Hicks, B. M. (2022). To vax or not to vax: Predictors of anti-vax attitudes and COVID-19 vaccine hesitancy prior to widespread vaccine availability. Plos one, 17(2), e0264019.